# Advances in Musculoskeletal Imaging in Juvenile Idiopathic Arthritis

**DOI:** 10.3390/biomedicines10102417

**Published:** 2022-09-27

**Authors:** Iwona Sudoł-Szopińska, Nele Herregods, Andrea S. Doria, Mihra S. Taljanovic, Piotr Gietka, Nikolay Tzaribachev, Andrea Sabine Klauser

**Affiliations:** 1Department of Radiology, National Institute of Geriatrics, Rheumatology and Rehabilitation, 02-637 Warsaw, Poland; 2Department of Radiology and Nuclear Medicine, Ghent University Hospital, C. Heymanslaan 10, 9000 Ghent, Belgium; 3Department of Diagnostic Imaging, The Hospital for Sick Children, University Avenue, Toronto, ON M5G 1X8, Canada; 4Department of Medical Imaging and Orthopaedic Surgery, University of Arizona, Tucson, AZ 85719, USA; 5Department of Radiology, University of New Mexico, Albuquerque, NM 87131, USA; 6Clinic of Paediatric Rheumatology, National Institute of Geriatrics, Rheumatology and Rehabilitation, 02-637 Warsaw, Poland; 7Pediatric Rheumatology Research Institute, Achtern Dieck 7, 24576 Bad Bramstedt, Germany; 8Rheumatology and Sports Imaging, Department of Radiology, Medical University Innsbruck, Anichstrasse 35, 6020 Innsbruck, Austria

**Keywords:** juvenile idiopathic arthritis, ultrasonography, elastography, magnetic resonance imaging

## Abstract

Over the past decade, imaging of inflammatory arthritis in juvenile arthropathies has significantly advanced due to technological improvements in the imaging modalities and elaboration of imaging recommendations and protocols through systematic international collaboration. This review presents the latest developments in ultrasound (US) and magnetic resonance imaging (MRI) of the peripheral and axial joints in juvenile idiopathic arthritis. In the field of US, the ultra-wideband and ultra-high-frequency transducers provide outstanding spatial resolution. The more sensitive Doppler options further improve the assessment and quantification of the vascularization of inflamed tissues, and shear wave elastography enables the diagnosis of tissue stiffness. Concerning MRI, substantial progress has been achieved due to technological improvements in combination with the development of semiquantitative scoring systems for the assessment of inflammation and the introduction of new definitions addressing the pediatric population. New solutions, such as superb microflow imaging, shear wave elastography, volume-interpolated breath-hold examination, and MRI-based synthetic computed tomography open new diagnostic possibilities and, at the same time, pose new challenges in terms of clinical applications and the interpretation of findings.

## 1. Introduction

Juvenile idiopathic arthritis (JIA) is the most common form of childhood arthritis [1,2,3,4,5,6]. It is a systemic disease in which the musculoskeletal system (MSK) is the most commonly affected. Joint involvement usually starts with synovitis and effusion, which are the main abnormalities in children for many years of disease, before the disease (mainly pannus) starts destroying hyaline cartilage, leading to bone erosions, joint destruction, and ankylosis [4,5]. The third most common pathology is tenosynovitis, which, in some JIA patients, may predominate over joint synovitis, or it can even be the only pathologic finding [7]. Contrary to adults, tenosynovitis rarely leads to tendon tears.

The clinical examination of joints is challenging in many cases, especially in the early stages of inflammation, and laboratory tests are not specific in the diagnostic process. Thus, there is increasing demand for novel imaging techniques to provide objective and accurate measures of inflammatory changes to monitor disease and treatment responses [1,8].

In 2022, the updated American College of Rheumatology Guidelines for JIA were published without significant modification in the part regarding imaging compared with the previous edition from 2013 [9]. Conventional radiography—according to recommendation—should be restricted to the assessment of JIA-associated damage or to investigate alternative diagnoses. Their use as a screening test prior to advanced imaging, for the purpose of identifying active synovitis or enthesitis, is strongly discouraged [9]. The role of ultrasound (US) is conditionally recommended for guidance of intraarticular injections or to localize the distribution of inflammation [9]. Indications for magnetic resonance imaging (MRI) are not mentioned. US or MRI are neither highlighted nor specified for various clinical indications in JIA, not only for synovitis and enthesitis, but also tenosynovitis, osteitis, and myositis.

Over the past decades, the imaging of inflammatory arthritis in juvenile arthropathies has significantly advanced due to technological improvements and the development of imaging recommendations, definitions, and protocols through systematic international collaboration. As a result, the approach to imaging inflammatory connective tissue diseases of childhood, including JIA, has changed over the last decade. Imaging modalities, such as US and MRI, have surpassed radiography in use for achieving an early diagnosis, assessing disease severity and prognosis, and monitoring of treatment efficacy [10].

The aim of this review was to summarize the latest advances in US and MRI and their practical applications in the diagnosis of MSK involvement in JIA based on the current literature.

## 2. Update in Ultrasound

MSK US is commonly performed in the pediatric population due to its lack of ionizing radiation and not requiring sedation, real-time dynamic imaging, and low cost. B-mode and Doppler US is well-suited for imaging peripheral joints and superficially located tendons, tendon sheaths, and entheses [11,12,13,14]. Therefore, US is especially useful in patients with JIA where the knee, hand, wrist, hip, ankle, and foot are among the most frequently affected joints [10,12,14]. US is also helpful in the differential diagnosis of JIA, which is based on clinical examination and the exclusion of other chronic arthropathies [12]. Nevertheless, apart from characteristic US findings, e.g., in subperiosteal abscess [15], the clinical scenario and laboratory data are crucial in differential diagnosis, which reflects the high sensitivity but low specificity of US [12,16,17,18,19].

In clinical practice, US is the primary technique used to detect and follow up on synovitis, tenosynovitis, bursitis, and enthesitis—the main pathologies in peripheral joints in JIA—having better sensitivity than clinical examination [12,20,21,22]. While synovitis and tenosynovitis are the main abnormalities in JIA, enthesitis is a rare pathology, mainly concerning the enthesitis-related arthritis (ERA) subtype of JIA. Contrary to adults, joint damage (erosions, cysts, and cartilage lesions/loss) is a late complication, and seen in only a small percentage of children, mainly with systemic and polyarticular subtypes of JIA that are commonly known to be the most clinically aggressive [10]. Contrary to polyarthritis, which is the most frequent pathology in adults with rheumatoid arthritis (RA), most frequently involving the hands and wrists, feet, and cervical spine, in JIA, the disease usually starts with monoarthritis, with the knee being the most frequently affected joint. The JIA process interferes with the development and growth of the skeleton, leading to the characteristic growth disturbances, even grow arrest, that are typical radiographic findings [10].

Prolonged tenosynovitis may lead to tendon involvement, however—contrary to adults with RA—in the pediatric population, full-thickness tears are rarely seen [10]. OMERACT defines enthesitis as a “hypoechoic and/or thickened insertion of the tendon close to the bone (within 2 mm from the bony cortex) which exhibits Doppler signal if active and that may show erosions, enthesophytes/calcifications as a sign of structural damage” [23]. This definition is used both for adult and pediatric patients with rheumatic diseases; it does not only lack the specificity, but it is also erroneously identified with primary inflammation, as all the previously mentioned elements are also present in other enthesopathies (i.e., overuse and post-traumatic). Thus, US does not enable differentiation between those clinical entities [10]. Moreover, whereas in adults, the entheses are avascular, vascularity of the tendinous (10–13 years of age) and peritendinous (4–9 years of age) regions were noted among healthy children, and it is considered a normal finding, indicating the need for caution while interpreting US findings [10].

Despite the above-mentioned limitations, many ongoing studies have been conducted in order to improve the US diagnosis of JIA.

In 2018, standardized procedures for US imaging in pediatric rheumatology developed by EULAR/PRES task force were published [24]. In 2019, the Outcome Measures in Rheumatology (OMERACT) redefined the definitions of US pathologies [25]. In contrast with the definition of synovitis in adults, the US definition in children also includes synovial effusion [25] (Table 1). In 2021, Sande et al., from the Oslo University Hospital, designed a more comprehensive semiquantitative scoring system for synovitis with a reference atlas in patients with JIA [26]. The novel scoring system proposes joint-specific scores for frequently affected joints in JIA and considers the variable US anatomy in the growing child that may lead to pitfalls when evaluating inflammation [26].

Contrary to synovitis, no JIA-specific scoring system is available for tenosynovitis, enthesitis, or bursitis in JIA.

According to the “European Federation of Societies for Ultrasound in Medicine and Biology (EFSUMB) Guidelines and Recommendations for Musculoskeletal Ultrasound”, US should be considered for fluid aspiration and guidance of intraarticular injections in the peripheral joints, tendon sheaths, or tendons [12]. US-guided injections of the spine and sacroiliac joints might be considered as an alternative to computed tomography or fluoroscopy guidance [12,27].

The advances in US include high-frequency multirow probes up to 70 MHz, with optimized transducers of 20 to 50 MHz for the assessment of MSK tissues, new technologies for microvascular imaging, and tissue elastography.

High-frequency and high-resolution US probes provide superior spatial resolution of superficial tissues, including the skin, and subcutaneous tissues. They are also beneficial for the assessment of small joints and low flow vessels, which make this technology suitable for microvascular imaging (Figure 1 and Figure 2).

Microvascular imaging, as with superb microvascular imaging (SMI), is a recent innovative Doppler technology. It is more sensitive than color Doppler US (CDUS) and power Doppler US (PDUS) [28,29] because it can visualize very-low-velocity flow (less than 1 cm/s) in microvessels with high resolution without the need to use a contrast medium [30,31]. Color SMI (cSMI) displays low-flow components in color overlaid on a grey-scale image with simultaneously high temporal and spatial resolution, and monochromatic SMI (mSMI) reveals the microvasculature with even higher sensitivity by subtracting the anatomical background [31] (Figure 2). The SMI algorithm provides flow quantification as the vascularity index (VI) by calculating the ratio of the color pixels to the total pixels [30]. The superiority of SMI to PDUS was recently described [28]. Clinical practice confirms that SMI readjusts the detection of vascularity within the inflamed synovium or tenosynovium and, as illustrated in Figure 1 and Figure 2, may significantly improve patient management and the stage of initial diagnosis and follow up. In terms of disease activity assessment, treatment monitoring, and remission setting, such information is particularly relevant. Early initial diagnosis and early detection of disease activity are crucial for achieving treatment response and reducing the disease sequelae [30]. Remission status is more accurately diagnosed by US than clinical examination; however, the prognostic value of subclinical synovitis is still being defined [12,32,33]. These issues became even more challenging given SMI sensitivity. One solution might be vascularity quantification with SMI, as mentioned above. In a study on 22 children with acute inflammation of the knee with JIA and 24 healthy pediatric volunteers, Unal et al. [30] found the VI by SMI and PDUS evaluated from supra-articular and infra-articular soft tissue could differentiate patients with JIA from volunteers.

Concerning the utilization of CDUS vs. PDUS for the assessment of tissue vascularity in arthritis, a recent study evaluated US sensitivity on different machines. The authors concluded that PDUS was more sensitive on half of the machines, whereas CDUS was more sensitive on the other half, using similar study settings [34]. Although this is a controversial topic, given the lower tendency for motion artifact with CDUS than with PDUS, CDUS has been more widely used for the assessment of vascularity of MSK disorders in pediatric healthcare centers than PDUS.

Another technology that has improved the diagnosis of JIA is elastography, which provides more in-depth evaluation of the muscles and superficial tissues.

Myositis and muscle wasting can occur in some patients with JIA [35,36,37,38,39,40]. Myositis may be triggered by biological treatment in JIA [35], and may be an element of JIA, a systemic subtype [36,37,38]. US has 82.9% sensitivity for detecting histologically proven myositis [11,39]. Inflamed muscles present increased echogenicity and may be thickened and/or swollen. In the chronic stage, muscle atrophy and progressive infiltration of fatty tissue are seen [11,40]. The advent of elastography, and especially shear wave elastography (SWE), has improved the diagnosis of myositis not only because it is less of an operator-dependent method but also by quantification of muscle stiffness. SWE is an objective, quantitative, and reproducible technique that uses an acoustic radiation force pulse sequence to generate shear waves, which propagate through the tissues perpendicular to the ultrasound beam, causing transient displacements [41]. SWE images are automatically coregistered with standard B-mode images to provide quantitative color elastograms with anatomic specificity. Shear waves propagate faster through stiffer and inflamed tissues, as well as along the long axes of tendon and muscle [41]. Bachasson et al. applied SWE to the biceps brachii of 34 patients with inclusion body myositis and found lower muscle stiffness in patients with more severe muscle weakness [42]. This suggests the potential of SWE to differentiate between degrees of disease activity. The effects of arthritis on biomechanical properties with softening of the distal third of the Achilles tendon [43] and patellar tendon [44] have been reported. Ulan [30] compared the elasticity of periarticular soft tissues (the quadriceps tendon, patellar tendon, and infra-articular and supra-articular soft tissues) via SWE between children with JIA and healthy children. They did not find any significant difference among patients with JIA and healthy subjects [30]. Because the development of fibrosis and contracture requires a long disease activity period, the differences in elasticity may be occult in early childhood [30].

## 3. Update in Magnetic Resonance Imaging

Magnetic resonance imaging (MRI) is a valuable technique for the assessment of pediatric musculoskeletal pathologies and is the most validated technique in JIA [45]. It is the most suitable technique for detecting synovial hypertrophy and bone marrow edema (BME) [1]. Compared with US, MRI has the advantages of allowing the three-dimensional evaluation of the peripheral and axial joints and the detection of inflammatory osteitis that cannot be visualized with radiography and US [46,47,48]. Additionally, MRI allows the evaluation of the complex or deep-seated joints.

Similar to US, the spectrum of inflammatory imaging findings detectable in the pediatric population is the same as in adults. Synovitis, tenosynovitis, bursitis, and enthesitis are the main findings on MRI in JIA. Inflammation also involves the bone, and MRI can easily detect bone marrow edema (BME) and osteitis, which cannot be assessed on US.

Recent years brought considerable improvements to the MRI diagnosis of both peripheral and axial arthritis in JIA. First was the implementation of whole-body MRI (WBMRI) for the screening of inflammation in JIA. Second, new MRI sequences have been introduced, such as 3D volumetric MRI and volume-interpolated breath-hold examination (VIBE). Third, due to breakthroughs in artificial intelligence (AI), new sequences have been developed to assess bony structures on MRI, such as from MRI-based synthetic CT (BoneMRI) images, in order to further improve the performance of MRI. Although advanced quantitative MRI techniques, such as dynamic contrast-enhanced (DCE)-MRI, T2 mapping, T1 rho, and diffusion-weighted imaging (DWI), are considered research techniques, they are increasingly being translated into daily practice.

In addition to the development of these new scanning techniques, substantial progress has also been made in the field of (semi)quantitative scoring systems for the assessment of inflammation in JIA. The knee is clinically the most commonly affected joint in JIA [48]. Pre- and standardized postcontrast sequences are warranted to accurately evaluate synovitis in the knee joint [45]. Recently, a pediatric-specific MRI scoring system for the knee (Juvenile Arthritis MRI Scoring (JAMRIS)) has been developed and validated [49,50,51], adding value to the tendency for the utilization of quantitative imaging. The hand and wrist are the second most common location of JIA involvement after the knee (Figure 3) [51]. In 2016, a group within the Health-e-Child (HeC) project and Outcome Measures in Rheumatology (OMERACT) MRI in JIA group published their recommendations for the MRI protocol of the wrist in JIA patients [52]. This protocol was initially based on the OMERACT Rheumatoid Arthritis MRI Scoring (RAMRIS) system for adults [53] and revised in the following years [45,54].

Hip and temporomandibular joints (TMJs) are the two dangerous locations of JIA because typically they are “silent” joints with mismatch between severity of symptoms and imaging findings and because their damage has a significant impact on the quality of life. The situation in which one of them is affected in isolation is most challenging for pediatricians who need to consider a long list the (so-called exclusion list) of other diseases in differential diagnosis. Ostrowska at al. [1] proposed a comprehensive scoring system for hip arthritis, including 24 active, chronic, and developmental hip lesions, and proposed the MRI summarized score, a sum of the scores of these lesions in an individual patient that showed 25% sensitivity and 100% specificity in discriminating hip arthritis from hip pain in juveniles without JIA (Figure 4).

For the ankle joint, the MRI summarized score did not allow the authors to distinguish between ankles affected by JIA and ankles affected by noninflammatory arthropathies [55]. The reduced spectrum of lesions seen on MRI with the predominance of BME over soft tissue abnormalities and the poor discriminatory value of MRI could result from the preselection of study patients, who may have been first referred for ankle US, and not further diagnosed, in case of a positive ultrasound US exam [55].

Arthritis involving the temporomandibular joint (TMJ) complicates 40–96% of cases of JIA, potentially leading to devastating changes to joint form and function [56]. Because of the complex nature of this joint, MRI has become the modality of choice for the assessment of early inflammatory changes as well as chronic abnormalities [45]. Both closed- and open-mouth views can be performed for the optimal assessment of the mandibular condyle morphology, and the disc position and function, although recent studies have suggested that only closed-mouth views suffice. MRI scoring systems for TMJ involvement by JIA are available [57,58,59]. MRI definitely improved patient management, whereas US, despite efforts to prove its diagnostic relevance [60,61], has lagged behind: not only is US insensitive to small amounts of effusion and synovitis, it does not provide any information on BME when it may be the only pathology. US can detect neither secondary osteoarthritis nor developmental disorders (condyle flattening, mandibular ramus thinning or shortening, abnormalities of the fovea, and articular eminence) nor disc morphology, and cannot evaluate its mobility (Figure 5).

Concerning the use of 1.5 vs. 3.0 T MRI for assessment of the TMJ, Inarejos Clemente et al. showed that inter-reader reliability and qualitative measures of image quality for assessment of TMJ more consistently improved with the coil offering higher resolution, and not increased magnet strength [62]. This study showed the value of both 1.5 and 3.0 T MRI for imaging small joints, such as the TMJ. Other technical factors (protocol, coil, etc.) play a substantial role in the final image quality.

In cervical spine arthritis in early detected JIA, subclinical involvement is often mandatory as well to prevent structural and irreversible damage. Even early ankylosis may develop [10,45,63] (Figure 6). The importance of MRI is undisputed because the majority of early lesions are reversible. Irreversible lesions seen both on MRI and radiography include erosions, dens deformations, subluxations, and ankylosis, as well as developmental complications including vertebral and disc hypoplasia [63,64,65,66]. Such complications are not rare; despite optimized diagnosis and treatment for JIA in the last years, they are diagnosed in up to 35% of JIA patients [67]. Radiography remains superior to MRI in the diagnosis of atlantoaxial and subaxial subluxations [67], whereas MRI can evaluate neural compression [65,66]. So far, an endorsed scoring system for MRI evaluation of the arthritis of the spine in children is not available.

Recent years brought significant improvements to the diagnosis of juvenile sacroiliitis. In the past, the diagnosis was based on radiography, which is not only insensitive but also potentially misleading [68]. Despite widespread consensus about the fact that MRI offers significant advantages for diagnostic evaluation of JIA in comparison with other imaging modalities and serves as an outcome tool for clinical trials, advances have been tempered by the lack of consensus-based definitions of the spectrum of lesions that may be observed on MRI [68]. Therefore, first, definitions for adults that were introduced in 2009 by the ASAS (Assessment of Spondyloarthritis International Society) MRI working group and were updated in 2019 have been used for children [69,70,71]. However, simply adopting adults’ definitions and scoring led to inaccurate diagnosis because of normal immature skeleton developmental appearance at MRI that may mimic pathologic changes [72,73,74,75]. In 2019, the first recommendations for the definitions of sacroiliac joint findings in JIA were published by the OMERACT Juvenile Idiopathic Arthritis MRI Working Group, an international consortium of rheumatologists and radiologists [76]. In 2021, the group developed and updated a semiquantitative MRI-based scoring system for the evaluation of sacroiliac joint (SIJ) inflammation and structural changes in children with JIA [77] (Table 2 and Table 3). Finally, the group published an atlas of MRI findings of juvenile sacroiliitis to illustrate the updated preliminary OMERACT pediatric JAMRIS (Juvenile Idiopathic Arthritis MRI Score) scoring system for active (part I) and structural lesions (part II) [68].

A hot topic and a point of recurrent discussion concerning MRI applications in JIA is the validity of using intravenous contrast in this pathology. Many studies point toward the increased sensitivity of contrast-enhanced MRI in the detection of synovitis and tenosynovitis in JIA. Hemke et al. [78] found unenhanced MRI of the knee joint to be of lower sensitivity (0.62) than gadolinium-enhanced MRI for the detection of synovial hypertrophy, but with retained high specificity (0.97). Dynamic contrast-enhanced MRI (DCE-MRI)-based quantification has been used for the early diagnosis and monitoring of treatment effect. It was found to be able to differentiate between active and inactive JIA in the knee and wrist based on the differences in signal intensity (SI) curve shapes [79]. The maximum enhancement values of the synovium in the wrist of JIA patients in remission seem to be able to predict clinical flares [80].

Concurrent with the existence of multiple studies corroborating the value of contrast-enhanced MRI, there are multiple publications on the accumulation of gadolinium contrast agent in the kidneys and brain (mostly in the basal ganglia), even after a prolonged period after contrast administration [81,82]. As JIA often involves numerous joints, the repetitive intravenous administration of gadolinium-based contrast agents can significantly burden the patient’s body [1]. Nusman et al. identified postgadolinium synovial enhancement in 52% of the knees of healthy children [83]; thus, a question remains regarding the value of injecting contrast in children. Ostrowska et al. showed that the noncontrast MRI of hip joints in JIA enables discrimination between JIA and not JIA in hip arthralgia [1].

As an alternative to contrast administration, new MRI techniques have been proposed, such as T2 mapping, double inversion recovery (DIR) techniques, and diffusion-weighted MRI (DWI), for the evaluation of synovitis in JIA [84,85].

Another advance that must be mentioned in this review is the availability of whole-body MRI (WBMRI) for the screening abnormalities in patients with JIA. WB-MRI has increasingly been used for the evaluation of rheumatologic diseases, enabling the assessment of the extent and activity of the disease involving the peripheral and axial joints, entheses, muscles, and bone marrow of the entire body in a single scanning session [86]. Thus, it overcomes the poor reliability of clinical examination, for example, of deep-seated joints such as the hip, sacroiliac, and temporomandibular joints, and provides a measure of the inflammatory load of the whole body [86]. Giraudo et al. conducted a survey in 2018 among radiologists, members of the European Society of Musculoskeletal Radiology, to identify the most common clinical indications and protocols used for WBMRI [87]. Inflammatory idiopathic myositis and chronic recurrent multifocal osteomyelitis (CRMO) were among the first indications for WB-MRI, followed by overlapping syndromes, JIA, other rheumatic disease, and nonspecific clinical signs of an inflammatory disease. In clinical scenarios not rarely included in a referral for WBMRI in pediatric patients, one can read: CRMO/JIA suspicion. This indicates the difficulties in making the final diagnosis of the rheumatologic disorder due to the lack of specific clinical and laboratory data for some pediatric rheumatologic conditions. Although in CRMO, mainly metaphyses and synchondroses are involved, and in JIA, epiphyses are mainly affected, there is considerable overlap between the imaging features of these two diseases (Figure 7).

In parallel with the scoring system for juvenile sacroiliitis, the aforementioned JAMRIS-SIJ score [85], in 2021, Panwar et al. [86] developed a standardized WBMRI scoring system to quantify the total inflammatory burden in children with JIA through formal consensus methods within an interdisciplinary group of experts. The developed scoring system includes 100 peripheral, 23 chest, and 76 axial joints, and 64 entheses, with 2–4 diagnostic parameters graded in each of the regions, using binary (presence or absence) and 2–3-level ordinal scores [86].

Concerning the imaging protocols for scanning SIJ, additional new MRI sequences, such as volume-interpolated breath-hold examination (VIBE) and MRI-based synthetic CT (BoneMRI), improve the performance of osseous abnormalities, such as erosions, sclerosis, and ankylosis (Figure 8 and Figure 9). VIBE is a 3D gradient echo (GRE) MRI sequence that has the advantage of higher spatial resolution, lower partial volume effects, and multiplanar reconstruction [46,88,89]. BoneMRI is a deep-learning-based technology generating CT-like images from three-dimensional T1-weighted multiple gradient echo sequences (3DT1MGEs) [90]. However, for the time being, this technique has not yet been validated for use in children.

In conclusion, continuous developments of US and MRI have improved the management of rheumatic patients. According to EFSUMB recommendations [12], US is more sensitive than clinical examination in the evaluation of inflamed joints and should be integrated into clinical examination in children with recent-onset inflammatory arthritis. It may also be considered for monitoring joint inflammation and to detect subclinical synovitis in JIA patients in clinical remission [12]. MRI is the method of choice to assess axial joints such as the TMJ and sacroiliac joints, as well as the soft tissue and osseous inflammation in all the joints that are hard to be evaluated by clinical examination and US, such as the hip, shoulder, and cervical spine. Advances in the hardware and software of US and MRI scanners have continuously increased our ability to make early diagnoses, follow up on the disease, and assess treatment response and clinical outcome [46]. Semiquantitative and quantitative assessments introduced in the recent years, along with specific definitions of lesions in the pediatric population, have allowed more objective evaluation and standardization of assessment [46].

In terms of US, the main advances in high-resolution probes and more sensitive technologies enabling slow vascular flow detection have been improving the detection and characterization of synovitis and tenosynovitis, whereas elastography provides a valuable adjunct to standard B-mode muscle imaging. Further research is needed to delineate the (minimal) discriminatory thresholds of diagnosed pathologies [25]. In terms of MRI, new semiquantitative scorings and validated protocols advance the MRI diagnosis of peripheral arthritis. Pioneering research has resulted in the introduction of child- and adolescent-specific definitions of sacroiliitis and the scoring of inflammatory lesions assessed on WBMRI. Newer MRI sequences, such as DWI, VIBE, and BoneMRI, have opened new possibilities to better diagnose and follow up patients with juvenile arthritis.

## Figures and Tables

**Figure 1 biomedicines-10-02417-f001:**
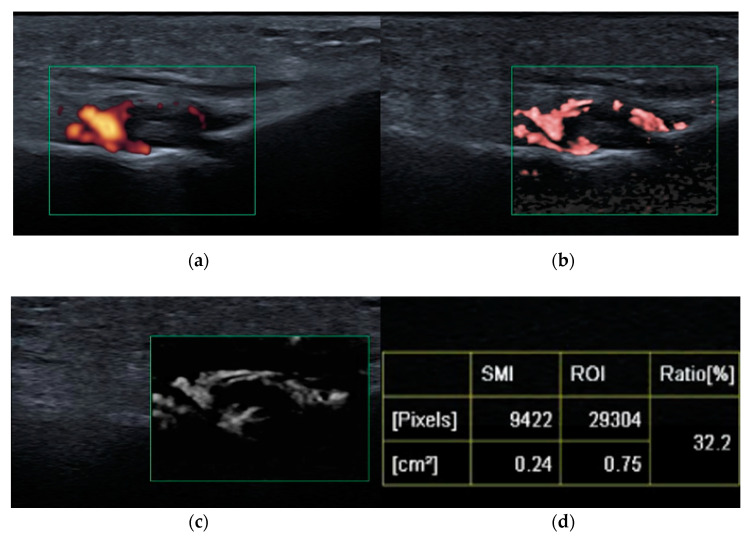
Third metatarso-phalangeal (MTP3, green square) joint synovitis shown on: power Doppler ultrasound (PDUS) (**a**), color superb microvascular imaging (SMI) (**b**), monochromatic SMI (**c**), and vascularity index (**d**); improved detection of hyperemia (vessels) achieved with SMI compared with PDUS.

**Figure 2 biomedicines-10-02417-f002:**
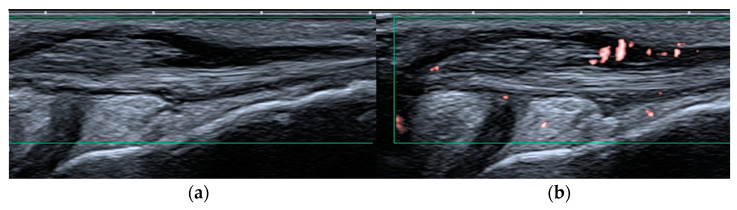
Peroneal tenosynovitis in a 6-year-old girl. Long-axis power Doppler ultrasound image (green rectangles) of the peroneal tendons (**a**) shows no hyperemia, which is evident on the more sensitive superb microvascular imaging ultrasound image (**b**), consistent with active tenosynovitis.

**Figure 3 biomedicines-10-02417-f003:**
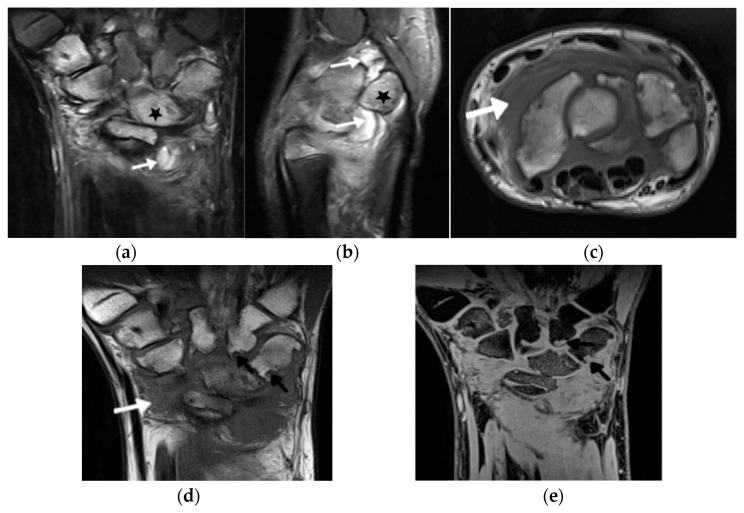
A 14-year-old boy with JIA and wrist arthritis. Coronal T2 Turbo inversion recovery magnitude (TIRM) (**a**), sagittal proton density fat saturated (**b**) and axial T1-weighted time spin echo (TSE) (**c**) MR images show synovitis (white arrows) and bone marrow edema (stars); coronal T1 TSE (**d**) and T1 volume-interpolated breath-hold examination (VIBE) (**e**) show numerous cyst-like changes and erosions (black arrows).

**Figure 4 biomedicines-10-02417-f004:**
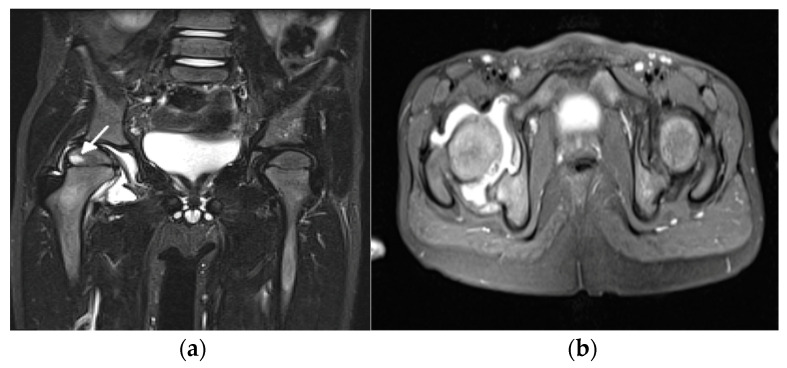
A 5-year-old girl with juvenile idiopathic arthritis with hip involvement. Coronal short tau inversion recovery (STIR) (**a**) and axial proton density (PD)-weighted fat saturated (**b**) MR images show right hip effusion and bone marrow edema in the right femoral epiphysis (arrow).

**Figure 5 biomedicines-10-02417-f005:**
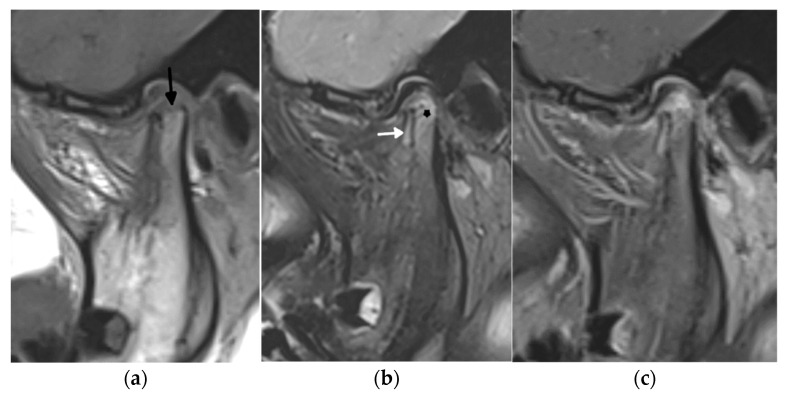
A 1.5 Tesla temporomandibular joint MRI with T1- (**a**), fat-saturated T2- (**b**), and postcontrast fat-saturated T1-weighted images (**c**) in a 7-year-old girl with juvenile idiopathic arthritis showing active and chronic inflammatory lesions: synovitis (white arrow), bone marrow edema in the mandibular condyle (star) in the upper part of the ramus, and erosions (black arrow).

**Figure 6 biomedicines-10-02417-f006:**
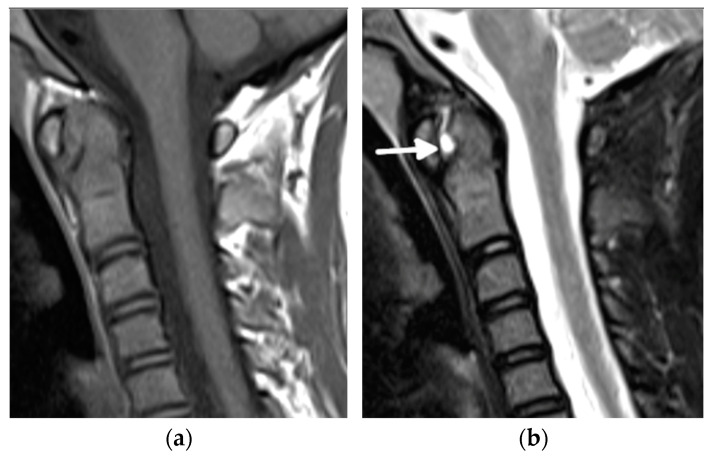
An 11-year-old old girl with juvenile idiopathic arthritis and cervical spine arthritis. Sagittal T1- (**a**), fat-saturated T2- (**b**), and postcontrast fat-saturated T1-weighted show prominent C1/2 joint arthritis with atlantodental joint fluid and synovial enhancement (arrows). Follow-up MRI with postcontrast fat-saturated T1-weighted sagittal MR image (**c**) 2 years later shows ankylosis of the atlanto-occipital joint, still with bone marrow edema consistent with ongoing active inflammation/osteitis (star) (**d**).

**Figure 7 biomedicines-10-02417-f007:**
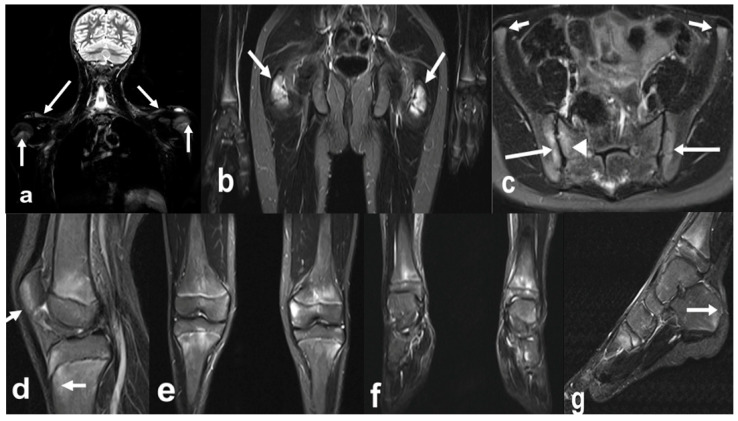
Whole-body MRI (WBMRI) in a 12-year-old boy with enthesitis-related arthritis (ERA) subtype of juvenile idiopathic arthritis (JIA). Superposition of findings of ERA and chronic recurrent multifocal osteomyelitis (CRMO) is seen on this WBMRI examination with STIR images of multiple body parts in multiple directions (**a**–**g**). Findings of CRMO are presented by ill-defined flamed-shaped areas of increased inversion recovery (IR) signal seen in the bilateral proximal humeral metaphyses and distal clavicles (**a**, arrows), distal femoral and proximal tibial metaphyses and at a lesser extent, epiphyses (**d**,**e**), distal tibial metaphyses and epiphyses, and tarsal and proximal metatarsals (**g**). Findings of JIA-related enthesopathies are represented by IR signal noted in the bilateral greater trochanters (**b**, arrows) at the site of insertion of gluteus medius and minimus, along the sacroiliac joints (**c**), iliac, long arrows and sacral aspects, arrowhead; (**c**), normal anterior–superior iliac spine apophyses short arrows), proximal and distal attachments of patellar tendon (**d**, arrows), and at the insertion of the Achilles tendon in the posterior calcaneus (**g**, arrow).

**Figure 8 biomedicines-10-02417-f008:**
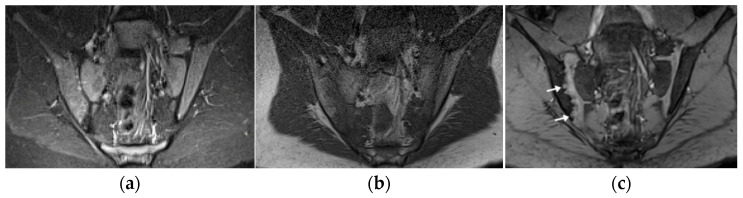
Follow-up MRI of the sacroiliac joints (only anterior superior part is shown) in a 14-year-old boy with juvenile spondyloarthritis (SpA), coronal short tau inversion recovery (STIR) (**a**), T1 TSE (**b**), and volume-interpolated breath-hold examination (VIBE) (**c**): joints without active lesions (**a**), with numerous erosions in the right joint that are difficult to depict on the T1, and are better visualized on the VIBE sequence (arrows).

**Figure 9 biomedicines-10-02417-f009:**
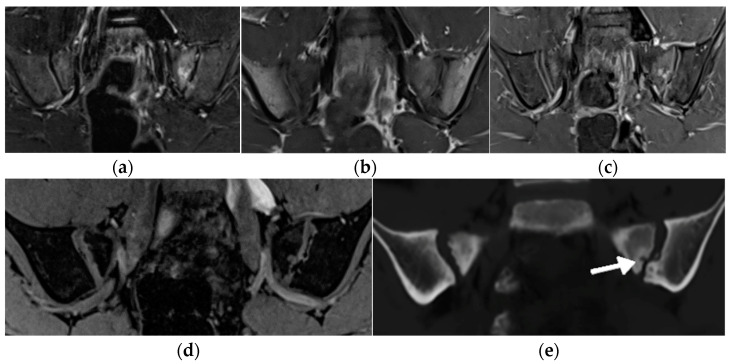
MRI of the sacroiliac joints of a 11-year-old boy with an active erosion in the left sacrum seen in consecutive sequences: T1-weighted time spin echo (TSE) (**a**), T2 short tau inversion recovery (STIR) (**b**), postgadolinium fat-saturated T1-weighted MR image (**c**), T1 volume-interpolated breath-hold examination (VIBE) (**d**), and BoneMRI (**e,** arrow).

**Table 1 biomedicines-10-02417-t001:** OMERACT combined scoring system for synovitis in children.

Grade	B Mode	Doppler
0	No signs of synovial effusion or synovial hypertrophy (i.e., no joint recess enlargement/capsular distension).	Absence of color/power Doppler signal within synovial hypertrophy with or without detection of normal physiological Doppler signals.
1 (mild)	Synovial effusion and/or synovial hypertrophy that leads to a mild change of joint recess appearance (i.e., mild joint recess enlargement/capsular distension).	Detection of up to 3 single Doppler signals within the area of synovial hypertrophy with or without normal physiological Doppler signals.
2 (moderate)	Synovial effusion and/or synovial hypertrophy that leads to a moderate change of joint recess appearance (i.e., moderate joint recess enlargement/capsular distension).	Detection of more than 3 single Doppler signals but less than 30% of the area of synovial hypertrophy with or without normal physiological Doppler signals.
3 (severe)	Synovial effusion and/or synovial hypertrophy that leads to a severe change of joint recess appearance (i.e., severe joint recess enlargement/capsular distension).	Detection of Doppler signals at more than 30% of area of synovial hypertrophy with or without normal physiological Doppler signals.

**Table 2 biomedicines-10-02417-t002:** Scoring system for assessing active inflammation of the sacroiliac joint by MRI according to the JAMRIS-SIJ scoring system.

Item	Definition	Score	
		Segmentation/Slice	Range/Slice
BME	An ill-defined area of high bone marrow signal intensity within the subchondral bone in the ilium or sacrum on fluid-sensitive images	4 quadrants/SIJ	0–8
BME Intensity	Presence of hyperintensity of the marrow on fluid-sensitive images using the signal of the presacral veins or cerebrospinal fluid (CSF) as reference	1 score/SIJ	0–2
BME depth	Continuing increased signal on fluid-sensitive images at depths ≥5 mm/≥1 cm from the articular surface	1 score/SIJ	0–2
Osteitis	An ill-defined area of high bone marrow signal intensity within the subchondral bone in the ilium or sacrum on contrast-enhanced T1-weighted sequences	4 quadrants/SIJ	0–8
Joint space fluid	High signal intensity equivalent to the CSF on fluid-sensitive sequences within the joint space of the cartilaginous portion of SIJ	halves/SIJ	0–4
Joint space enhancement	Increased signal intensity on contrast-enhanced T1-weighted sequences within the joint space of the cartilaginous portion of SIJ	halves/SIJ	0–4
Inflammation in erosion cavity	Increased signal intensity on fluid-sensitive or contrast-enhanced T1-weighted sequences in an erosion cavity of cartilaginous portion of SIJ	halves/SIJ	0–4
Enthesitis	Increased signal intensity in bone marrow and/or adjacent soft tissue on fluid-sensitive or contrast-enhanced T1-weighted sequences at sites where ligaments and tendons attach to a bone, excluding retroarticular enthesitis	score per case	0–1
Capsulitis	Increased signal on fluid-sensitive or contrast-enhanced T1-weighted sequences involving superior portion of SIJ capsule	superior halves/SIJ	0–2

**Table 3 biomedicines-10-02417-t003:** Scoring system for assessing structural damage of the sacroiliac joint by MRI according to the JAMRIS-SIJ scoring system.

Item	Definition	Score	
		Segmentation/Slice	Range/Slice
Sclerosis	A substantially wider than normal area of very low bone marrow signal intensity within the subchondral bone in the ilium or sacrum on a nonfat suppressed sequence, preferably a nonfat suppressed T1-weighted sequence; this feature must also be present on all other sequences, as available	4 quadrants/SIJ	0–8
Erosions	A focal loss of low signal of cortical bone at osteochondral interface and adjacent marrow matrix on T1-weighted images	4 quadrants/SIJ	0–8
Fat lesion/Fat metaplasia	Homogeneous increased signal intensity within subchondral bone marrow on T1-weighted images	4 quadrants/SIJ	0–8
Backfill	A high signal on non-contrast-enhanced T1-weighted sequences in a typical location for erosion, with signal intensity greater than normal bone marrow, clearly demarcated from adjacent bone marrow by an irregular band of low signal, reflecting sclerosis at border of original erosion	halves/SIJ	0–4
Ankylosis	Presence of signal equivalent to regional bone marrow continuously bridging a portion of the joint space between iliac and sacral bones	halves/SIJ	0–4

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
