# Peer review of "Advances in Musculoskeletal Imaging in Juvenile Idiopathic Arthritis"

_biomedicines, 2022, doi:10.3390/biomedicines10102417_

Round 1

Reviewer 1 Report

The review Advances in Musculoskeletal Imaging in Juvenile Idiopathic Arthritis by Iwona Sudol-Szopinska, Nele Herregods, Andrea S. Doria, Mihra S. Taljanovic, Piotr Gietka, Nikolay Tzaribachev and Andrea S. Klauser is an attractive and elaborate work that updates the technological advances of the musculoskeletal imaging modalities, i.e., the latest developments in ultrasonography (US) and magnetic resonance imaging (MRI) in juvenile idiopathic arthritis (JIA), which are very important in this childhood pathology for correct diagnosis, the visualization of the biological effects of the treatment (outcome) and the progression of the disease.

Featured as advances in US imaging are the latest ultra-wideband and ultra-high frequency transducers to provide outstanding spatial resolution, as well as more sensitive Doppler solutions to further improve the assessment and quantification of inflamed tissue vascularization, or the shear wave elastography, which allows the diagnosis of tissue stiffness.

Regarding MRI, the well-grounded progress is presented by the authors and highlighted as new solutions, such as superb microflow imaging (SMI), shear wave elastography, volume-interpolated breath-hold examination, and MRI-based synthetic computed tomography, which open new diagnostic possibilities and, at the same time, pose new challenges in terms of clinical applications and interpretation of findings, especially for children diagnosed with JIA.

The results are accurately presented and discussed in detail. All figures presenting medical imaging in JIA are sharp and well explained. The final conclusions are pertinent.

The manuscript is not overloaded with unnecessary information and offers an overview of the progress in the musculoskeletal medical imaging well and specifically applied in JIA.

The article has several shortcomings as follows:

The title is not written correctly and contains a final unnecessary dot.

The keywords: The authors did not adhere to MeSH for choosing keywords and have several imperfections in keyword wording and their form.

References are not written in the required MDPI / Biomedicines format and do not contain the Digital Object Identifier (DOI), to uniquely identify the article or the document, and to provide it with a permanent web address (URL). All references must be double-checked.

The comparative presentation of all new musculoskeletal imaging techniques presented in this review but drawn in a final outline (with all pluses, minuses, limitations and main applications), or as a Graphical Abstract, would greatly improve the quality of this review.

A list of abbreviations MUST be completed and must be carefully reviewed.

Overall, I congratulate the authors for preparing this article, which required a lot of work, and the result is impressive.

More English and style issues could be improved by a native English speaker at a final reading on your MDPI platform.

Author Response

Reviewer 1.

The title is not written correctly and contains a final unnecessary dot. 

Answer: This has been corrected, thank you.

The keywords: The authors did not adhere to MeSH for choosing keywords and have several imperfections in keyword wording and their form.

Answer: This has been corrected, thank you.

References are not written in the required MDPI / Biomedicines format and do not contain the Digital Object Identifier (DOI), to uniquely identify the article or the document, and to provide it with a permanent web address (URL). All references must be double-checked.

Answer: This has been corrected, thank you.

The comparative presentation of all new musculoskeletal imaging techniques presented in this review but drawn in a final outline (with all pluses, minuses, limitations and main applications), or as a Graphical Abstract, would greatly improve the quality of this review.

Answer: Conclusion of this manuscript has been corrected, thank you.

A list of abbreviations MUST be completed and must be carefully reviewed.

Answer: This has been done, thank you.

Reviewer 2 Report

I read with interest this review by Sudol-Szopinska on the potential applications of MSK imaging in JIA. 

The authors provide a nice illustrative overwiev on the main advances of imaging in such condition. 

The paper needs to be re-structured as the way how different sections of the review are organised is confusing and it does not seem to follow a logical flow. For instance, in the ultrasound section, the authors first describe SMI, then myositis/muscle wasting and myositis again. After that, they talk about Doppler again (Colour vs power Doppler), to conclude on the potential applications of SWE in myositis? A similar lack of logical flow is observable in the MRI section-

The OMERACT definitions cited in the paper refer to adults (23). While the differences in the definition of synovitis have been pointed out by the authors, there is no mention on how the scoring of enthesitis, tenosynovitis and bursitis is done in patients with JIA. 

I would introduce a brief paragraph (both in the US and MRI sections) describing the spectrum of normal imaging findings detectable in paediatric populations and how that differs from adults. 

I cannot see any "a", "b", "c", and "d" in Figure 1 and it is not clear to what these correspond. In addition, please do not include pictures with presence of PD below the bone indicating possible artifacts. 

Author Response

Reviewer 2.

The paper needs to be re-structured as the way how different sections of the review are organised is confusing and it does not seem to follow a logical flow. For instance, in the ultrasound section, the authors first describe SMI, then myositis/muscle wasting and myositis again. After that, they talk about Doppler again (Colour vs power Doppler), to conclude on the potential applications of SWE in myositis? A similar lack of logical flow is observable in the MRI section

Answer: This has been corrected, for both parts, concerning US and MRI. Thank you.

The OMERACT definitions cited in the paper refer to adults. While the differences in the definition of synovitis have been pointed out by the authors, there is no mention on how the scoring of enthesitis, tenosynovitis and bursitis is done in patients with JIA. 

Answer: This has been explained, thank you.

I would introduce a brief paragraph (both in the US and MRI sections) describing the spectrum of normal imaging findings detectable in paediatric populations and how that differs from adults.

Answer: This has been added, thank you.

I cannot see any "a", "b", "c", and "d" in Figure 1 and it is not clear to what these correspond. In addition, please do not include pictures with presence of PD below the bone indicating possible artifacts. 

Answer: This has been corrected, thank you.

Reviewer 3 Report

The thesis organizes and presents routine imaging procedures in an up-to-date manner.

Makes a recommendation for clinical incorporation. With the help of US and MRI, they want to provide a more accurate diagnosis and better treatment.

It neatly organizes current soft tissue, bone and joint imaging. Children's results were used in the thesis.

Several questions need to be clarified in the thesis.

This should be treated as a summary study. Why didn't they increase the number of items and make a comparative study of the test methods?

Why was 3T MRI not used? What is the reason that the picture is also suitable for 1.5 T?

This raises several problems.

How would you like to implement this in the hospital's protocol?

I consider the thesis valuable.

Author Response

Reviewer 3

This should be treated as a summary study. Why didn't they increase the number of items and make a comparative study of the test methods?

Answer: As it was written in the Introduction, the aim of this study was to present advances in imaging in JIA and provide an overview, not to compare the existing methods. Latest technological and research achievements were described and illustrated.

Why was 3T MRI not used? What is the reason that the picture is also suitable for 1.5 T? This raises several problems. How would you like to implement this in the hospital's protocol?

Answer: 1,5 Tesla MRIs are most commonly used in daily practice in many centers and provide high quality images. However, 3T MRI may be perfectly used for all the discussed indications and scoring systems. Care must be taken only, since 3T MRI machines are very sensitive to motion artifacts so not ideal for use in young patients who have difficulty in lying still.

All presented imaging techniques of this manuscript have been implemented in clinical practice, although in some cases they are available only with modern machines, such in a case of ultrasound – SME and SWE. MRI is more and more commonly used, becoming a method of choice to diagnose juvenile sacroiliitis.

The use of MRI has continued to increase in recent years. In adults it increased from 62 to 139 per 1000 person-years in the U.S. and from 13 to 89 per 1000 person-years in Canada (Ontario) between 2000 and 2016. In children in the same period of time, it increased from 9 to 21 per 1000 person-years in the U.S., and from 4 to 16 per 1000 person-years in Ontario [1]. In Europe the availability of MRI scanners relative to the number of inhabitants was highest in Finland (with 2.9 MRI units per 100 000 inhabitants), Cyprus (2.0), Italy (1.8) and Sweden (1.7) [2], Despite huge disparities in the availability of MRI scanners across countries, e.g. Japan being 400 times higher than in Myanmar, which are associated with the gross domestic product per capita of the country [3], the usage of MRI is steadily growing in the entire world and becoming less restrictive to specialized healthcare facilities. This tool is becoming an essential part of medicine through the use of non-invasive processes for accurate diagnosis and determination of disease status, response to therapy and prognosis. 

Concerning the Reviewer’s question on the utilization of 1.5 vs 3.0 T MRI for assessment of temporomanidbular joints (TMJs) Inarejos Clemente et al. showed that inter-reader reliability and qualitative measures of image quality for assessment of TMJs improved more consistently with the coil offering the higher resolution, rather than increased magnet strength [4]. Other technical factors (protocol, coil, etc) play a substantial role on the final quality of images.

WBMRI is not popular approach, comparing to single joint MRI, despite many advantages. It is especially helpful for the whole body screening for inflammatory features in both, soft tissues and bones. 

References:

  1. Smith-Bindman R, Kwan ML, Marlow EC, et al. Trends in Use of Medical Imaging in US Health Care Systems and in Ontario, Canada, 2000-2016. JAMA2019;322:843–856.

  1. https://ec.europa.eu/eurostat/web/products-eurostat-news/-/ddn-20210702-2. Last access, September 10, 2022.

  1. Watari T, Hlaing TM, Kanda H. The Choosing Wisely Initiative and MRIs: Over- and Under-Diagnosis in Japan and Myanmar. Cureus 2021 7;13(4):e14342.

  1. Inarejos Clemente EJ, Tolend M, Junhasavasdikul T, Stimec J, Tzaribachev N, Koos B, Spiegel L, Moineddin R, Doria AS. Qualitative and semi-quantitative assessment of temporomandibular joint MRI protocols for juvenile idiopathic arthritis at 1.5 and 3.0T. Eur J Radiol. 2018 Jan;98:90-99.

Round 2

Reviewer 2 Report

Many thanks for addressing my comments